# Ruthenium(II)-catalysed remote C–H alkylations as a versatile platform to *meta*-decorated arenes

Jie Li[1,*], Korkit Korvorapun[1,*], Suman De Sarkar[1,*], Torben Rogge[1], David J. Burns[1], Svenja Warratz[1] & Lutz Ackermann[1]

The full control of positional selectivity is of prime importance in C–H activation technology. Chelation assistance served as the stimulus for the development of a plethora of *ortho*-selective arene functionalizations. In sharp contrast, *meta*-selective C–H functionalizations continue to be scarce, with all ruthenium-catalysed transformations currently requiring difficult to remove or modify nitrogen-containing heterocycles. Herein, we describe a unifying concept to access a wealth of *meta*-decorated arenes by a unique arene ligand effect in proximity-induced ruthenium(II) C–H activation catalysis. The transformative nature of our strategy is mirrored by providing a step-economical entry to a range of *meta*-substituted arenes, including ketones, acids, amines and phenols—key structural motifs in crop protection, material sciences, medicinal chemistry and pharmaceutical industries.

[1] Institut für Organische und Biomolekulare Chemie, Georg-August-Universität Göttingen, Tammannstraße 2, 37077 Göttingen, Germany. * These authors contributed equally to this work. Correspondence and requests for materials should be addressed to L.A. (email: Lutz.Ackermann@chemie.uni-goettingen.de).

The functionalization of otherwise inert C–H bonds by means of transition metal catalysis has emerged as an increasingly powerful platform in organic synthesis, with transformative applications to medicinal chemistry, material sciences and drug design[1–10]. Since the substrates of interest display a variety of C–H bonds with close dissociation energies, achieving positional selectivity in intermolecular C–H transformations is paramount[11–15]. Thus, chelation assistance has proven particularly instrumental for proximity-induced ortho-C–H functionalizations[16–19]. In stark contrast, remote arene functionalizations continue to be challenging, with major recent progress being achieved by inter alia complementary palladium[20–29], iridium[30,31], rhodium[32] and ruthenium[33–39]

catalysis through steric control, template assistance, weak hydrogen bonding, transient mediators or catalytic σ-activation by ortho-C–H metalation (Fig. 1a)[40]. Despite undisputable advances, these methods typically offer access to only a single compound class. Furthermore, all protocols for ruthenium-catalysed meta-C–H functionalization[33–39] continue to be restricted to nitrogen-containing heterocycles, such as 2-arylpyridines, as the directing group. Since such heteroarenes are difficult to modify or remove[39,41], the synthetic utility of this strategy is significantly compromised. Within our program on sustainable C–H activation, we have now addressed these major limitations in C–H activation technology by developing remote imine C–H functionalizations by a unique arene ligand effect,

**Figure 1 | Transformative ruthenium(II)-catalysed meta-C–H functionalization regime.** (**a**) Previous reports: selectivity control by (1) steric interactions, (2) template auxiliaries, (3) hydrogen bonding, (4) transient mediator and (5) difficult to remove or modify pyridines. (**b**) Unifying concept to a wealth of meta-decorated arenes.

## Table 1 | Reaction development for *meta*-selective C–H functionalization.

| Entry | Ligand | Solvent | Yield (%) |
|-------|--------|---------|-----------|
| 1 | MesCO$_2$H | 1,4-dioxane | 30 |
| 2 | 1-AdCO$_2$H (**4**) | 1,4-dioxane | 52 |
| 3 | 1-AdCO$_2$H (**4**) | PhH | 54 |
| 4 | 1-AdCO$_2$H(**4**) | PhMe | 58 |
| **5** | **1-AdCO$_2$H (4)** | **PhCMe$_3$** | **73, 76*** |
| 6 | Piv-Val-OH | PhMe | 17 |
| 7 | Boc-Val-OH | PhMe | 26 |
| 8 | Boc-Ile-OH | PhMe | 28 |
| 9 | Piv-Ile-OH (**5**) | PhMe | 33 |
| 10 | Piv-Ile-OH (**5**) | PhCF$_3$ | 41 |
| **11** | **Piv-Ile-OH (5)** | **PhCMe$_3$** | **64** |

*[RuCl$_2$(*p*-cymene)]$_2$ (2.5 mol %), 1-AdCO$_2$H (15 mol %). TMP = 3,4,5-trimethoxyphenyl. Reaction conditions: **1a** (0.5 mmol), **2a** (1.5 mmol), [RuCl$_2$(*p*-cymene)]$_2$ (5.0 mol %), ligand (30 mol %), K$_2$CO$_3$ (1.0 mmol), solvent (2.0 ml), 120 °C, 20 h, yield of isolated products. Bold entries indicate optimal ligands (4 and 5), solvent (PhCMe$_3$) and corresponding yields.

unleashing the full potential of C–H activation technology. Our approach is characterized by an expedient substrate scope, providing a broad access to various *meta*-decorated arenes, including synthetically meaningful ketones, alcohols, amines and acids, that constitute integral structural motifs in material sciences, crop protection and drug design (Fig. 1b)[42,43]. Notable features of our findings are not limited to (1) a remarkable arene ligand effect in ruthenium C–H activation chemistry, (2) a distinct catalyst design and (3) a tandem multicatalysis[44] approach involving both remote *meta*- and *ortho*-C–H functionalization with the aid of a single ruthenium(II) catalyst manifold.

## Results

**Development of *meta*-C–H alkylation.** We commenced our studies by probing the effect exerted by carboxylates and solvents on the challenging *meta*-C–H alkylation of synthetically useful ketimines **1** (Table 1 and Supplementary Table 1). Sterically congested 1-AdCO$_2$H (**4**)[45] was found to be an efficient ligand for the desired remote C–H functionalization process (entries 1 and 2). Notably, among a variety of solvents, *tert*-butylbenzene set the stage for a particularly effective *meta*-C–H functionalization catalysis that strongly contrasts to the previously employed 1,4-dioxane and toluene solvents (entries 1–5). Given the power of mono-protected amino acids (MPAAs) in C–H activation[35,46,47], we also explored different MPAAs in the *meta*-C–H functionalization process (entries 6–9), with Piv-Ile-OH (**5**) emerging as the best in class (entries 9–11).

**Substrate scope.** The versatility of the optimized ruthenium(II)-catalysed *meta*-C–H alkylation was explored with substituted ketimines **1** and tertiary bromides **2**, initially employing the ruthenium(II) catalyst derived from the MPAA Piv-Ile-OH (**5**). The catalytic system was found to be versatile, yet the ruthenium(II) biscarboxylate catalyst generally proved more powerful (Fig. 2). We were pleased to observe that both tertiary and secondary alkyl bromides **2** were compatible electrophiles

in the carboxylate-assisted ruthenium(II)-catalysed *meta*-C–H functionalization. A range of electronically differentiated ketimines **1** performed well under the optimized reaction conditions with both cyclic and acyclic tertiary alkyl bromides **2**. It is noteworthy that the alkyl bromide **2e** containing a highly reactive alkyl chloride motif furnished the desired product **3ae** with excellent levels of chemoselectivity. The remarkable versatility of the optimized ruthenium catalyst was reflected by fully tolerating synthetically valuable functional groups, such as chlorides, heteroarenes, ester, ketones, thioethers or amines, within intramolecular as well as intermolecular competition experiments, including a robustness screen[48] protocol (see the Supplementary Table 3). Propiophenone-derived ketimine **1e-f** underwent the *meta*-cycloheptylation to selectively deliver the desired products **3**, while an aldimine substrate gave thus far only less satisfactory yields of 20%. Likewise, the naphthalene derivative **1l** furnished *meta*-substituted arene **3lf–3lm** as the sole products by positional selective C–H functionalization, while the structurally complex steroid **3of** could be prepared by remote C–H activation. It is noteworthy that the corresponding 3,4,5-trimethoxyphenyl (TMP)-amine could be recovered after its traceless removal in high yields (see **3eb**). Moreover, synthetically useful Lewis-basic heterocycles, such as morpholine, pyran and piperidine, were fully accepted by the robust ruthenium(II) catalysis regime.

**Mechanistic considerations.** Given the unique efficacy of the ketimine-assisted *meta*-C–H functionalization by ruthenium(II) catalysis, along with the unconventional solvent effect, we became attracted to delineating its mode of action. To this end, intra- and intermolecular competition experiments revealed the *meta*-C–H alkylation to exclusively occur on the more electron-deficient aromatic moieties (Fig. 3, and the Supplementary Figs 1 and 2), with the geometric isomers of substrate **1r** undergoing facile interconversion even at ambient temperature (Supplementary Fig. 3). It is noteworthy that these observations strongly contrast with the trend previously observed in *meta*-sulfonylations[38] and alkylations[37] of 2-phenylpyridines, in which electron-rich arenes usually reacted preferentially. In contrast to previous proposals[38,49], our findings thus render an electrophilic substitution manifold unlikely to be operative here.

Furthermore, the use of typical radical scavengers (Fig. 4a), enantiomerically enriched substrate **2m** (Fig. 4b) and the diastereomerically pure alkyl halides **2p** provided strong support for a radical-based mechanism (Fig. 4c).

Detailed kinetic experiments with mono-metallic catalyst [Ru(O$_2$CAd)$_2$(*p*-cymene)] (**6**) highlighted a first-order dependence with respect to both the single-component catalyst **6** and the ketimine **1a** (Fig. 5), with saturation kinetics being observed for the alkyl bromide **2a** (see the Supplementary Fig. 4). An Arrhenius plot analysis highlighted an activation barrier of 99 kJ mol$^{-1}$. To rationalize the unique effect exerted by the aromatic solvent *tert*-butylbenzene, we independently prepared the novel single-component complex **7**. It is noteworthy that the well-defined catalyst **7** featured a significantly reduced induction period, along with an overall improved robustness and catalytic efficacy, indicating a unique arene ligand effect in ruthenium-catalysed C–H activation catalysis.

**Late-stage diversification.** The outstanding synthetic utility of the remote[50] imine C–H functionalization approach for late-stage diversification of the thus obtained *meta*-alkylated arenes was reflected by operationally simple transformations in a user-friendly one-pot fashion (Fig. 6). Facile reduction of the ketimines **8**, hence, provided valuable benzyl amine derivatives **9**.

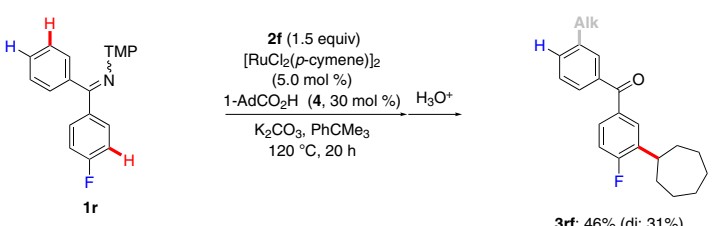

**Figure 2 | Substrate scope.** Versatility of the ruthenium(II)-catalysed *meta*-alkylation.

**Figure 3 | Intramolecular competition experiment.** Alkylation occurs on the more electron-deficient aromatic group.

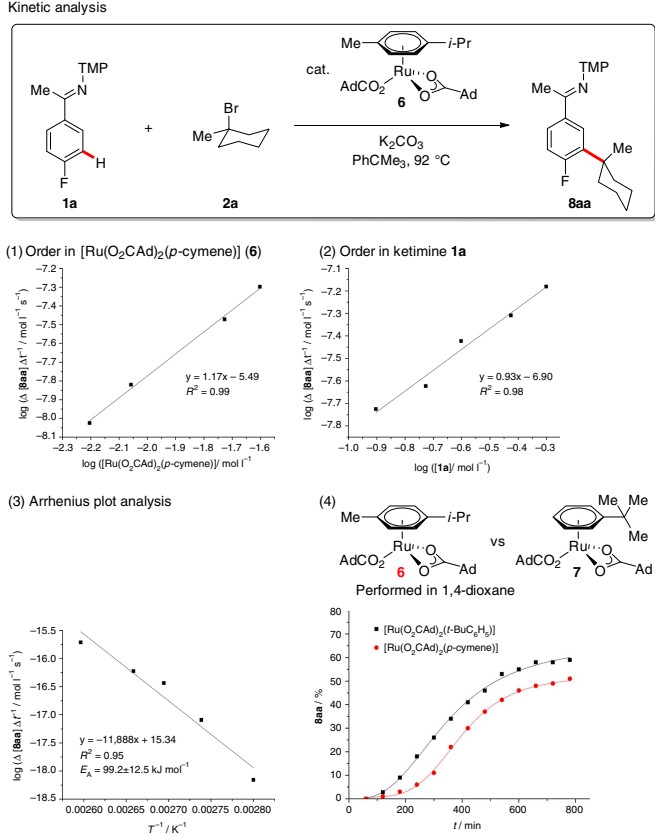

**Figure 4 | Mechanistic studies.** Probing a radical-based mechanism by (**a**) the addition of radical scavengers (**b**) the use of enantiomerically enriched alkyl halide **2m** and (**c**) the use of diasteromerically pure **2p**.

**Figure 5 | Kinetic analysis.** Order in (1) catalyst **6** and (2) reagent **1a**, for detailed information, see the Supplementary Information. (3) Arrhenius plot analysis. (4) Comparison of performance with single-component ruthenium(II) arene catalysts **6** and **7**.

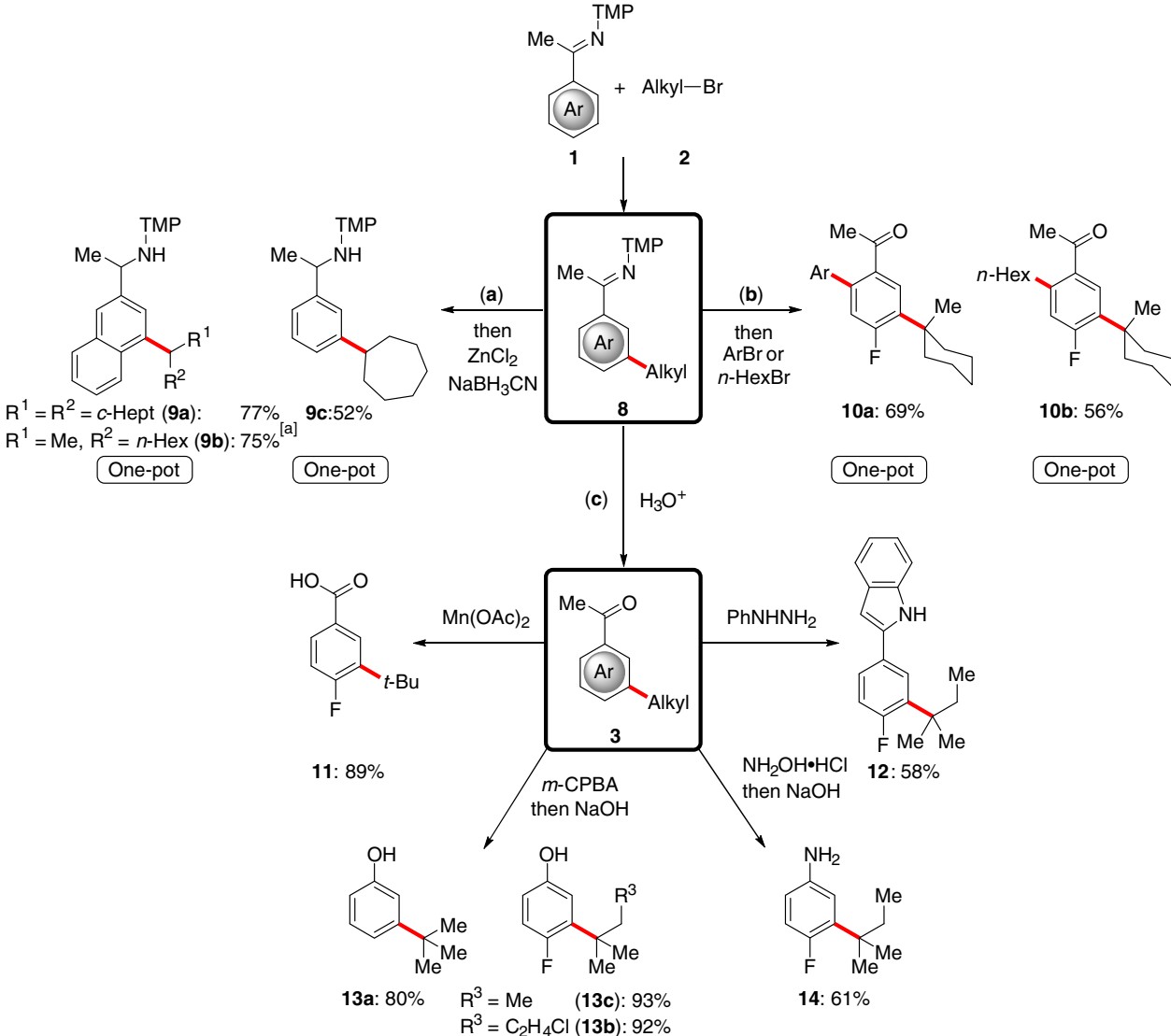

**Figure 6 | Late-stage diversification.** The *meta*-C–H functionalization of ketimines **1** as transformative platform into synthetically meaningful and biologically significant compounds. For detailed information, see the Supplementary Information. (**a**) One-pot remote-C–H functionalization/reduction. [a]$dr = 1.0{:}1.2$. (**b**) One-pot *meta*-C–H alkylation and *ortho*-C–H arylation/alkylation regime. (**c**) Late-stage diversification to access acids **11**, indoles **12**, phenols **13** and anilines **14**. Ar = 4-MeOC$_6$H$_4$.

Gratifyingly, sequential *meta*-C–H alkylation followed by *ortho*-C–H arylation or alkylation provided access to densely substituted aromatics **10** with no additional catalyst being required, showcasing the enabling power of our approach within a user-friendly multicatalysis regime. The unique synthetic versatility of the *meta*-substituted arenes **3** was further illustrated by transformative diversifications (Fig. 6), forming useful building blocks and biologically significant motifs, such as carboxylic acids **11**, and indoles **12**. In this regard, the preparation of *meta*-substituted phenols **13** and anilines **14** is particularly noteworthy, since classical methods of organic synthesis, such as the Friedel–Crafts reactions, fall short in providing access to the *meta*-decorated scaffolds due to the substrate's inherent bias for *ortho/para*-guided selectivity.

## Discussion

In summary, we have presented a versatile concept for the step-economical preparation of *meta*-substituted arenes by remote C–H functionalization. Henceforth, a considerable arene

ligand effect set the stage for a powerful ruthenium(II) catalysis manifold that expedited efficient secondary and tertiary C–H alkylations of easily accessible ketimines with exceptional positional selectivity. Operationally simple one-pot protocols delivered synthetically useful *meta*-functionalized benzyl amines, while multicatalytic C–H functionalizations produced densely *meta*-/*ortho*-substituted arenes within a one-pot process. The transformative nature of the approach was highlighted by the preparation of a wealth of *meta*-substituted arenes, including ketones, amines, indoles, acids and phenols.

## Methods

**General techniques.** Catalytic reactions were performed under a N$_2$ atmosphere using pre-dried glassware and standard Schlenk techniques. 1,4-Dioxane was dried over sodium and freshly distilled under N$_2$. Yields refer to isolated compounds, estimated to be >95% pure as determined by [1]H-nuclear magnetic resonance ([1]H-NMR) and gas chromatography. Thin-layer chromatography was performed on Merck, TLC Silica Gel 60 F$_{254}$ with detection under ultraviolet light at 254 nm. Chromatographic separations were carried out on Merck Geduran SI-60 (0.040–0.063 mm, 230–400 mesh ASTM). Infrared spectra were recorded on a Bruker FT-IR alpha-P device. Electron ionization mass spectrometry was recorded

on Jeol AccuTOF at 70 eV; electrospray ionization mass spectrometry was recorded on Bruker Daltonik micrOTOF and maXis and LIFDI with a Linden CMS. Elemental analyses were measured on an Elementar Vario EL 3 analyser. Melting points were measured on Stuart melting point apparatus SMP3; values are uncorrected. NMR spectroscopy was performed at 300, 400 or 500 MHz ($^1$H-NMR), 75, 100 or 125 MHz ($^{13}$C-NMR, APT), 282, 376 or 470 MHz ($^{19}$F-NMR) and 282 or 376 MHz ($^{19}$F{$^1$H}) on Bruker Avance III HD 300, Avance III 300, Avance III 400, Avance III HD 500, Varian Unity-300, Inova 500 and Inova 600 instruments. If not otherwise specified, chemical shifts ($\delta$) are provided in p.p.m. and spectra referred to non-deuterated solvent signal. Analytical high-performance liquid chromatography analysis was performed on Agilent 1260 Infinity equipped with Daicel CHIRALPAK IC-3 (4.6 mm × 250 mm, 3 μm particle size, 1 ml min$^{-1}$ flow rate). Optical rotary power was measured on Jasco P-2000 polarimeter as a 0.04 g per 100 ml solution in MeOH at 589 nm and 23.0 °C. For NMR spectra of all products in this article, see the Supplementary Figs 5–60.

**General procedure for catalysed *meta*-C–H alkylation.** Ketimine 1 (0.50 mmol) [RuCl$_2$(*p*-cymene)]$_2$ (15.3 mg, 25.0 μmol), 1-AdCO$_2$H (27.3 mg, 0.15 mmol) and K$_2$CO$_3$ (138 mg, 1.00 mmol) were placed in a pre-dried 25 ml pressure tube. The reaction tube was then evacuated and backfilled with nitrogen three times. Alkyl bromide 2 (1.50 mmol) and PhCMe$_3$ (2.0 ml) were added and the mixture was stirred at 120 °C for 20 h. At ambient temperature, HCl (2 N, 3.0 ml) was added, and the resulting mixture was stirred for an additional 3 h, and extracted at ambient temperature with EtOAc or Et$_2$O (3 × 20 ml). The combined organic layers were dried over Na$_2$SO$_4$ and concentrated *in vacuo*. Purification of the residue by column chromatography (*n*-hexane/EtOAc or *n*-pentane/Et$_2$O) yielded phenone 3.

**Data availability.** The authors declare that the data supporting the findings of this study are available within the article and its Supplementary Information files. All data are also available from the authors on reasonable request.

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

## Acknowledgements

Generous support by the Alexander von Humboldt Foundation (fellowship to S.D.S.), the Chinese Scholarship Program (fellowship to J.L.), the DAAD (fellowship to K.K.), the DFG (SPP 1807) and the European Research Council under the European Community's Seventh Framework Program (FP7 2007–2013)/ERC Grant Agreement No. 307535 is gratefully acknowledged.

## Author contributions

J.L. developed the remote *meta*-C–H alkylation. S.D.S., K.K. and D.J.B. explored the substrate scope and the arene ligand effect. T.R. and S.W. conducted the mechanistic studies and T.R. performed the robustness test. L.A. conceived and supervised the project. L.A. wrote the manuscript.

## Additional information

**Competing interests:** The authors declare no competing financial interests.

