## [Peer review file · Nature Communications]

REVIEWERS' COMMENTS:

Reviewer #2 (Remarks to the Author):

The "revised" manuscript is on the whole significantly better than the original and I can recommend publication. With that said, there are some points that could probably be addressed to improve the manuscript, and while they are not essential, they would provide more useful information than is currently presented.

For instance compound 8 is shown to give 9a and 9b by reduction. But the diastereoselectivity of the reduction is not explored since the only "alkyl" group in 8 is one lacking any stereocenters. I assume the d.r. is 1:1 if 3l-m is reacted under the same conditions but perhaps the authors can comment?

In classifying various strategies to make meta substituted derivatives the authors do not mention the strategy first reported by Catellani DOI: 10.1055/s-2003-37102 and subsequently reported in DOI: 10.1021/ol051628n wherein meta substituted compounds are produced without the need for a directing group, but requiring a halide which is subsequently reduced. They may want to add these references or comment on this alternative approach, to be complete.

We wish to thank the reviewers for their insightful suggestions to improve the quality of our manuscript. We have addressed all comments as follows:

Reviewer 2:

- 1) We have performed the suggested experiments. The new results are described in the modified Figure 6.
- 2) The suggested citations were added as new references [49,50].